[Supplementary Material]



# Appendix

## A    Proof of Lemma and Theorem

### A.1    Proof of Lemma 1

*Proof.* Let $B_{jk}$ denote $\mathrm{Bal}(b_j, b_k)$. Since $0 \leq b_k \leq 1$ (as $S = \sum_k b_k + K$), we have $0 \leq B_{jk} \leq 1$. In addition, $B_{jk} = 1$, if $b_j = b_k \neq 0$; $B_{jk} = 0$, if $b_j b_k = 0$. Thus, we have $\sum_{j \neq k} b_j B_{jk} \leq \sum_{j \neq k} b_j$, where the equality holds when $B_{jk} = 1, \forall j$. Therefore, we have

$$diss(\omega) = \sum_{k=1}^{K} b_k \left[ \frac{\sum_{j \neq k} b_j B_{jk}}{\sum_{j \neq k} b_j} \right] \tag{13}$$

$$\leq \sum_{k=1}^{K} b_k \overset{(a)}{=} \frac{\sum_{k=1}^{K} r_k}{S} \overset{(b)}{=} \frac{S - K}{S} = 1 - \frac{1}{C} \tag{14}$$

where (a) is due to the definition of $b_k$ in equation 5 and (b) is due to the summation constraint in equation 1 and $W = K$. $\square$

### A.2    Proof of Lemma 2

*Proof.* Using the definition of uncertainty mass in equation 5 and substituting $W$ by $K$, we have

$$0 \leq vac(\omega) = \frac{K}{S} = \frac{K}{\sum_{k=1} r_k + K} \leq 1 \tag{15}$$

where equality is achieved when $\sum_{k=1}^{K} r_k = 0$. $\square$

### A.3    Proof of Theorem 1

*Proof.* For (1), ($\Rightarrow$) is easy to show as $S = K$ implies $\sum_{k=1}^{K} r_k = 0$ and $vac(\omega_{\mathrm{y}^*}) = 1$; for ($\Leftarrow$), using equation 2 and non-informative base rates, we have $P(\mathrm{y}^* = k) = 1/K, \forall k$, which achieves a maximum $H[\mathrm{y}^*]$ as $\log K$.

For (2), we first prove ($\Rightarrow$). For $\mathrm{y}^* = \arg\max H[\mathrm{y}]$, we have $P(\mathrm{y}^* = k) = 1/K, \forall k$. Thus, $(r_k + a_k K)/S = 1/K, \forall k$. For $S \to \infty$, denote $S = CK$ and we have $r_k/S + a_k/C = 1/K, \forall k$. Let $C \to \infty$, we have $r_k/S \to 1/K, \forall k$. Thus, we have $\mathrm{y}^* = \arg\max diss(\omega_{\mathrm{y}})$ due to Corollary 1. To prove ($\Leftarrow$), $diss(\omega_{\mathrm{y}*}) = 1$ implies that $r_1 = ... r_k ... = r_K$ and $S \to \infty$. Hence, $\lim_{S \to \infty} P(\mathrm{y}^* = k) = \lim_{S \to \infty} (r_k + a_k K)/S = 1/K$, which implies that $\mathrm{y}^* = \arg\max H[\mathrm{y}]$ for $S \to \infty$. $\square$

## B    Experimental Settings

We choose the Adam optimizer to train ADL for 600 epochs and setting the learning rate to 0.001. The coefficient of evidence strength loss, $\lambda_1$, is set to 0.005 (cross validated from $\{0.001, 0.005, 0.03, 0.05\}$). The coefficient of the $L_2$ regularizer, $\lambda_2$, is set to 0.05 (cross validated from $\{0.001, 0.005, 0.01, 0.03, 0.05, 0.08\}$). The $\lambda$ for anchor sample identification is set to 0.005 (cross validated $\{0.001, 0.005, 0.03, 0.05\}$). We choose RBF as our kernel function with length scale set to 1 (cross validated from $\{0.01, 0.1, 1\}$).

## C    Ablation Study

We have conducted a detailed ablation study to clearly demonstrate the effectiveness of each major technical components. We adopt the MNIST data with 5 missing classes for illustration purpose.

**Sampling function.** Figure 6(a) compares proposed sampling method with other different sampling criteria: entropy, vacuity only, and dissonance only. The result confirms the effectiveness of the dynamically balanced sampling method. It is interesting to see that using vacuity alone performs quite well in the initial phase but only converges to a lower accuracy in the end. In contrast, using dissonance is slow to start but able to converge to a higher accuracy. The entropy curve roughly stays in the middle of the above two curves.

Figure 6: Ablation study (a) Comparison of different sampling criteria; (b) Comparison with different fixed vacuity/dissonance weighting; (c) Comparison with random anchor selection, no anchor samples, and attention kernel; (d) Impact of the characteristic length scale

Figure 7: Samples with a high vacuity (top) and high dissonance (bottom) in early and late AL iterations, respectively.

**Vacuity/dissonance balancing.** Figure 6(b) shows the results using different fixed vacuity/dissonance ratios. The dynamically balanced sampling method clearly outperforms the fixed weighting. This also demonstrates the usefulness of the proposed entropy decomposition theory. Since the sampling goal of AL changes with the accumulation of the labeled data, the optimal AL behavior can only be achieved by adaptively adjust the importance of vacuity and dissonance in the sampling function.

**Anchor sample identification.** We have compared ADL with the randomly selected anchor samples from unlabeled data and not using any anchor samples in Figure 6(c). ADL clearly outperforms random selection, which in turn performs better than not using any anchor samples. We further compare with the attention kernel as a more advanced distance metric for anchor sample identification. The attention kernel is the major component in the matching network [19], where the spatial invariance is ensured by CNN and the dimensionality of the inputs is reduced through two correlated LSTM projections. However, the attention kernel (our current implementation) is much slower to compute as compared with our approach especially when facing a very large unlabeled pool as the entire candidate data samples need to be embedded every iteration when the training/testing data are changed along with AL. Thus, if the improvement is not significant (see Figure 6(c)) and when the efficiency becomes a bottleneck for a large unlabeled pool, the proposed approach appears to be a good choice as it can provide a good balance between quality and efficiency, which is critical for AL.

We also implemented a state-of-the-art OOD detection method [20] based on Mahalanobis distance for anchor sample identification. In general, the ODD detection methods primarily focus on detecting OOD data samples so they may return isolated noisy samples, which will not help active sampling. In contrast, our approach aims to locate anchor samples from densely distributed areas in the unlabeled data space to better support active sampling. Our results in Figure 6(c) show that the method in [20] performs very close to random anchor sampling (worse than ours). A potential reason may be that this is a supervised model that requires well-labeled data to train, which is not the case for AL, especially in early stage.

**RBF length scale.** We have investigated the impact of the characteristic length scale used in RBF kernel on AL performance. Figure 6(d) shows that the ADL model performance is fairly robust to the length scale and only shows minor change with different choices.

**Sample Images Chosen by ADL.** In order to better demonstrate the effectiveness of the vacuity measurement, we start active learning with 5 classes omitted from the initial training. Figure 7 (top) shows the samples with highest vacuity selected by ADL in the first 30 AL iterations. The first four of them are from missing classes. This clearly demonstrates the effectiveness of using vacuity to explore the data space. As a result, data samples from the missing classes are quickly identified and being labeled. The last sample is from class '3', whose examples have already been exposed to ADL. However, the writing style of this sample is very different than other instances from the same class, which result in a high vacuity. Figure 7 (bottom) shows the samples with highest dissonance selected by ADL in the last 100 AL iterations. By observing their predicted belief mass, we find that the high dissonance is due to the conflicting belief over multiple classes. For example, the first sample is confusing between classes '4' and '6'; the second sample is confusing among classes '5','6', and '8'; the third sample is confusing among classes '4','7', and '9'; the fourth sample is confusing between classes '4' and '6'; and the fifth sample is confusing between classes '0' and '9'.

## D  Additional Experimental and Comparison Results

**Additional Results.** We obtain similar AL curves for notMNIST and CIFAR-10 when starting AL with 7 and 8 classes as with 5 and 6 classes, which are shown in Figure 8. In Figure 9, we also report the AL performance on the three datasets when there is no missing class. ADL still achieves the best performance in all cases with slightly less advantage than other models.

Figure 8: AL performance on notMNIST and CIFAR-10 (start with 7 and 8 classes)

Figure 9: AL performance with no missing classes

Figure 10: Comparison with batch-mode deep AL models

**Comparison with batch-mode AL models.** For batch AL, a key requirement is to ensure the diversity of the samples in the batch. This is the focus some recent state-of-the-art deep learning based batch-model AL models, through diverse gradient embedding [16] and cover set [3], respectively. Our model performs single batch AL, using second-order uncertainty to choose the single best sample each iteration. Further, the batch AL model usually uses a large batch size (100-1000), more than all labels ADL collects. We did experiment to compare ADL with [16] and [3] and the results are shown in Figure 10. ADL is clearly better in single batch AL and also better in small batches of 2-4. At batch 6, [16] outperforms ADL, which is still better than [3].

**Source code**

The code for this work can be found in https://drive.google.com/drive/folders/1vRRvKlD_sZMM9by5Fne6gvKqf3ZyYzUm?usp=sharing