[Reviews · NeurIPS 2020]

Review 1

Summary and Contributions: The authors propose to do active learning by using uncertainty estimates, derived from a Dirichlet Prior Network model, decomposed into vacuity and dissonance to do active learning in a low-data regime. The proposed approach outperforms competing methods.

Strengths: The current work is relevant to the NeurIPS community. The proposed approach is rooted in firmly established Dempster-Shafer theory and Dirichlet Prior Network models, and outperforms alternative methods. The paper is clear and well written. Also, I believe this is the first paper, that I've seen at least, which examines active learning using a Dirichlet Prior Network style model.

Weaknesses: I have several concerns regarding this work. 1. Firstly, I'm not entirely convinced by the need to introduce an evidence-based Dempster Schafer / Subjective logic framework. The proposed decomposition into vacuity and dissonance is essentially the same thing as decomposing into epistemic uncertainty and aleatoric uncertainty. Why not consider the tractable closed form mutual information decomposition into total, epistemic and aleatoric uncertainties which was derived in previous work? I believe that decomposition would, broadly speaking, have many of the same quantities. Similarly, the definition for vacuity (W/S) is almost the same as for expected pairwise KL-divergence introduce in Malinin et al ( (W-1)/S ). Given a DPN trained with the proposed loss, I suspect that mutual information (vacuity) and expected entropy (dissonance) would enable many of the same behaviours. Furthermore, it would allow comparing directly to approaches such as BALD (which just use mutual information, derived via dropout). 2. It would have been really good to see active learning experiments of a greater scale, using for example SVHN/CIFAR-100/TinyImageNet. While the proposed method outperforms the baselines in this small-scale setting (MNIST/notMNIST/CIFAR-10), it is important to see how well this scales to larger and more challenging datasets, and therefore active learning scenarios. I am also surprised by your choice of architecture. Would there be difficulties in getting it to work on more modern architectures, such as WideResNets and DenseNets?

Correctness: The proposed method seems to be mathematically correctly, and is rooted in firmly established theory.

Clarity: Paper is clear and well written.

Relation to Prior Work: Yes.

Reproducibility: Yes

Additional Feedback:


Review 2

Summary and Contributions: This paper offers a new take on data uncertainty under the guise of evidence-based uncertainty which it decomposes into vacuity and dissonance, under the framework of subjective logic (ref). This decomposition enables the design of a vacuity-aware regularizer that is claimed to guide the model convergence towards the correct decision boundaries, even in over-parametrized models with little data. In the context of active learning, the authors propose a sampling function that is the weighted sum of dissonance and vacuity where vacuity is annealed for the purpose of “balancing” as the authors call it.

Strengths: > The evidence-based entropy decomposition, while not novel or surprising, streamlines and formalizes the quantification of vacuity and dissonance that are important to the practical deployment of decision making, especially in the context of out-of-distribution test data. > The proposed decomposition allows the design of a vacuity-aware regularizer, based on a set of "anchor samples" which represent OOD areas in the data space. Optimizing this loss ensures the model predicts high vacuity (lack of evidence) in these areas. This is reminiscent of aspects of contrastive learning which as been found to help the calibration of uncertainty estimation. > A weighted sum of vacuity and dissonance is used as a new sampling function for active learning. The effect of dissonance is increased over training iterations on the assumption that the decision boundary is properly avoiding OOD areas and can be fine-tuned on high data density regions. > Empirical evaluation and ablation studies seem to support the validity of this approach as it outperforms the baselines.

Weaknesses: > Writing: > > The manuscript is quite verbose and the writing is awkward in certain parts. Everything prior to the experimental section could be rewritten in a more concise matter and certainly with less repetition. > > It should be made clear what "evidence-based entropy" is especially when claiming that entropy can be decomposed into vacuity and dissonance. While it is clear that those two quantities affect entropy, it is not clear how that happens analytically. A reader can be misled to think that entropy is equal to the sum of dissonance and vacuity given the abundant repetition of the phrase "entropy decomposition into vacuity and dissonance" throughout the text. > This work is not well placed with respect to prior literature in that it omits work on data uncertainty such as Malinin & Gales, 2018 and Hafner et al., 2018. > Empirical evaluation: I would have preferred to see more details in the main text instead of the appendix. This should be possible if the first 6 pages are shrunk down to 4. -----------POST-REBUTTAL UPDATE----------------- While the feedback has addressed some of my concerns, I still find both the lacking empirical evidence and the verbose presentation of this work's novelty work (as well as its placement with regards to prior work) insufficient to increase this score beyond a "marginally above the acceptance threshold".

Correctness: The claims and method are generally correct.

Clarity: The clarity of this paper is lacking. The writing is verbose and at times repetitive.

Relation to Prior Work: Differences to previous contributions are not well discussed as this papers omits prior work as Noise-Contrastive Priors (Hafner et al., 2018) and Prior Networks (Malinin & Gales, 2018) which tackle data uncertainty. Calling data uncertainty evidence-based uncertainty obfuscates such connections and might confuse the reader. In fact, noise-contrastive priors leverage the idea of training a model to output high uncertainty for data points outside of the training distribution. This is similar to the vacuity regularization term based on anchor samples. On the other hand, prior networks analyze the effects of vacuity and dissonance (without calling them as such) on OOD detection.

Reproducibility: Yes

Additional Feedback:


Review 3

Summary and Contributions: The paper proposes an uncertainty-aware active learning strategy for a deep learner inspired by subjective logic. Similar to prior networks, the deep learner is designed to regress evidence for a Dirichlet distribution such that areas far from the labeled training data will exhibit high vacuity and areas near class label boundaries will exhibit high dissonance. The active learning part is annealed so that it initially looks for high vacuity samples to label and over time switches to searching for high dissonance samples to label.

Strengths: The idea to incorporate vacuity and dissonance in an active learning framework seems highly novel and potentially very useful. The actual design of the evidence-based deep learner in how it biases high vacuity for samples far from labeled samples is very clever. The experiments do demonstrate the utility of the approach against BALD and a strawman uncertainty-aware approach derived from EDL.

Weaknesses: EDL does not necessarily exhibit high vacuity (or epistemic uncertainty) far from the labeled training data as it is not trained to do so. It seems that a more fitting uncertainty-aware strawman to compare against is prior networks (see ref [11] in the paper). Once the proposed deep learner is trained, it seems that the deep leaner is no longer uncertainty (or evidential) aware. Certainly, the experiments only demonstrate the classification accuracy of the deep learner and not its ability to properly capture vacuity. [After reading the Author Feedback] The authors have addressed my main concerns. If possible, it would be nice to see the Prior Network results in the paper.

Correctness: From what I can tell all the equations are correct. There might be some conditions where dissonance is undefined, e.g., when u = 1.

Clarity: The paper is well written, but there are some points that are not clear. It seems that after each sample is labeled, the parameters for the evidential deep model must be updated. Some words about the computational complexity for this update seems necessary.

Relation to Prior Work: The paper does an adequate job discussing prior related work.

Reproducibility: Yes

Additional Feedback: Of minor note: Line 97: Should DU be EU? Around eqn. (5) it would help the reader to understand what is a reasonable value for W. Line 189: K -> \infty, should K actually be C? Line 277: It seems that Beta can go negative. Is that true?

[Author Response · NeurIPS 2020]

We thank all reviewers for their valuable feedback and constructive suggestions. Major comments are addressed below.

**Reply to Reviewer 1**

*Q1: Need for an evidence-based subjective logic (SL) framework.* The SL framework provides the theoret-
ical underpinning to perform a principled, fine-grained analysis between the 1st-order uncertainty (i.e., pre-
dictive entropy as the total uncertainty) and 2nd-order uncertainty (vacuity + dissonance), where evidence
plays a central role to unveil the underlying (dynamic) relationship among different uncertainties. Under-
standing this dynamic relationship is essential to derive a theoretically sound data sampling process in AL.
In particular, Theorem 1 shows the total uncertainty dynamically shifts between

MNIST: start with 5 classes

high vacuity and high dissonance as more evidence is collected. Guided by this
theory, AL can be regarded as an evidence collection process. The evidence-based
uncertainties (i.e., vacuity + dissonance) derived under SL, offer a natural way
to determine the sources of uncertainty during AL, which starts by focusing
on vacuity in early stage when evidence is limited and then gradually shifts
to dissonance. Since vacuity and dissonance both depend on evidence, using
evidence provides a principled way to trace the dynamic shift between these
two sources of uncertainty to best guide data sampling in AL. This is the **key**
**advantage** over other types of uncertainty, such as epistemic and aleatoric, which
only focus a certain aspect of uncertainty, and their (dynamic) relationship is
hard to be precisely quantified as in vacuity and dissonance. Thus, using these
uncertainty measures lacks the capability to dynamically adjust the sampling
process as the nature and focus of uncertainty change when more data samples
are labeled. In sum, while there are various forms of uncertainty measures, the evidence-based (2nd-order) uncertainty,
i.e., vacuity + dissonance, offers the most suitable way for active sampling, as justified by our theory and empirical
evaluation. The right figure includes additional comparison with other uncertainties: epistemic, aleatoric (Kendall &
Gal, 2017), and distributional uncertainty of prior networks (Malinin & Gales, 2018). As can be seen, ADL converges
much faster than other uncertainty based sampling functions, which empirically confirms its effectiveness in AL. We
will report the comparison results on all other datasets in the revised paper.

*Q2: Choice of architecture.* We choose a relatively simple architecture to demonstrate that the good AL performance is
due to the sampling function instead of a strong classifier. Several works (eg, [7] and [11]) follow a similar rationale.

*Q3: Experiments of a greater scale.* We thank the reviewer for suggesting these large-scale image datasets. Limited by
time, we were not able to conduct the experiments on these large datasets and collect the active learning results. An
interesting future direction is to combine the architectures suggested by the reviewer and our sampling function and
apply to these large datasets.

**Reply to Reviewer 3**

*Q1: What "evidence-based entropy" is when claiming entropy can be decomposed into vacuity and dissonance.* Thank
you for the suggestion. Entropy decomposition means a high entropy dynamically shifts between a high vacuity and a
high dissonance as more evidence is collected, instead of a simple sum of these two uncertainties. We will make this
clear in the revised paper.

*Q2: Relation to prior work.* The prior networks model (Malinin & Gales, 2018) proposes distributional uncertainty
(DU) for OOD detection. While DU can be regarded as a type of epistemic uncertainty that can be used for data
sampling in AL, the prior network needs to be properly trained as its parameter must encapsulate knowledge of both
in-domain distribution and the decision boundary, making it not very suitable for AL. This is also evidenced by our
additional comparison result in Fig. (a). The Noise-Contrastive Priors (Hafner et al. 2018) can also be used for OOD
detection as it encourages high uncertainty near the boundary of the training data. However, in the initial phase of AL
when the training data is very limited, this measure can be insufficient to explore data samples faraway from the training
data. R1 (Q1) offers a deeper discussion on why vacuity + dissonance is more effective for AL than other uncertainties.

**Reply to Reviewer 4**

*Q1: Compare to prior networks...ability to capture vacuity.* For comparison with prior networks, please refer to Fig.
(a) and R3 (Q2). As the model continues to be trained with AL, it remains uncertainty aware but the total uncertainty
(especially vacuity) will decrease. Fig.3 (page 7) shows how vacuity changes along with AL.

*Q2: Parameter update and complexity.* ADL is retrained once a labeled sample is added. Time complexity is
$O(training\_size \times epoch)$, which is efficient given the small training size in AL.

*Q3: Minor note.* DU should be EU; $W = K$ is commonly used for a non-informative prior; it should be $C \to \infty$; $\beta$
should be non-negative, which is ensured in our experiments. We will fix these typos.

[Meta-Review · NeurIPS 2020]

Incorporating notions of "vacuity" and "dissonance" in active learning is novel, and reviewers are convinced of its improvements. The empirical evaluation regarding certain baselines and scale are a bit lacking, and while the work may be sufficient for acceptance, the reviews all believe the work would at least benefit from further discussion to related work in the paper revision: Prior Networks and Noise Contrastive Priors. In addition, promoting more discussion on the need for Dempster-Schafer as R1 asks, is very welcome. I congratulate the authors!